# Language and family entrepreneurship: empirical research based on micro survey data of CFPS in China

Chenglin Ren [1*¤], Yijia Tang[2¤]

1 School of accounting, Chongqing Technology and Business University, Chongqing, China, 2 Investment Banking Department of Chongqing Branch, China ZheShang Bank, Chongqing, China

¤Current address: School of accounting, Chongqing Technology and Business University, Chongqing, China.

* renchenglin@ctbu.edu.cn

## Abstract

Language serves as a vital link between individuals. In a multi-ethnic country like China, significant differences exist between Standard Mandarin and various regional dialects, which can influence family entrepreneurs' access to entrepreneurial resources and information. This paper examines the theoretical and empirical impact of Standard Mandarin and dialects on family entrepreneurship choice and performance, using microdata from the China Family Panel Studies (CFPS). The results show that families who predominantly use Standard Mandarin have a higher probability of starting a business, with cognitive competence and social networks as two important channels through which language influences the likelihood of entrepreneurship. Specifically, the effect of Standard Mandarin on urban families' entrepreneurship choice is more significant, whereas the impact on rural families is not as pronounced. Regardless of whether they are in urban or rural areas, speaking a dialect is more beneficial for entrepreneurial families in terms of integrating into the local community, leading to better entrepreneurial performance. Further analysis reveals that in most of the ten dialect regions of China, Standard Mandarin has a significant positive impact on family entrepreneurship choice, but a significant negative impact on family entrepreneurial performance. In high-income, young, eastern, and central region families, the probability of starting businesses is higher for families using Standard Mandarin, while dialects have a significant positive impact on family entrepreneurial performance in most sub-samples.

## Introduction

In recent years, China's economy has undergone a critical period of transformation, shifting from quantitative to qualitative growth. However, the new economic growth targets and structural imbalances between supply and demand in the labor market

**Data availability statement:** Data and code are held in OPENICPSR with the DOI https://doi.org/10.3886/E204821V1.

**Funding:** This study was supported by Chongqing Technology and Business University in the form of a grant [project no. KFJJ2018031] to CR. The funders had no role in study design, data collection and analysis, decision to publish, or preparation of the manuscript.

**Competing interests:** The authors have declared that no competing interests exist.

during this transition have led to a challenging employment situation. Previous studies have demonstrated that micro businesses initiated by family entrepreneurs provide the majority of jobs in middle-income areas and are becoming a sustained driver of the economy [1–4] in. In response to the changes in the current employment landscape, the Chinese government has implemented measures to promote entrepreneurship and street vending activities. These initiatives aim to encourage more individuals to leverage their family resources to start businesses and alleviate employment pressures [5–8].

Current research on the influencing factors of family entrepreneurship has focused on two main perspectives. One perspective emphasizes the role of human capital in family entrepreneurship.. Studies have shown that the personal qualities of family entrepreneurs are among the most important factors influencing entrepreneurship. For example, Xiong and Li have found that improvements in cognitive ability significantly promote entrepreneurial willingness and revenue, with cognitive competence reflecting differences in human capital among entrepreneurs [9]. Yan has measured the differences in human capital among rural family entrepreneurs based on factors such as years of education, financial knowledge, and health condition, concluding that the more years of education, the greater the incremental probability of family startups will be [10]. Yin et al. have also found that an extension of financial knowledge significantly promotes proactive participation in family entrepreneurship [6]. The other perspective focuses on the formation of social capital among family entrepreneurs, particularly for those who lack collateral and are less preferred by large financial institutions [11]. Su et al. note that social networks can relieve financial constraints on family entrepreneurs through private lending, which has a significant positive effect on rural family entrepreneurship [12]. Zhang et al. have found that social networks can broaden information channels for families, enhancing the entrepreneurial income levels of both urban and rural families, with a more pronounced effect in urban settings [13]. Notably, previous studies have highlighted that the linguistic abilities of family entrepreneurs have a significant influence on the acquisition of both human and social capital [14–17].

Language acts as a crucial channel for information exchange. This is particularly evident in China, a country with diverse ethnic groups and a multitude of dialects, where language serves as a "stepping stone" for individuals to integrate into different communities. According to the Language Atlas of China, the country has ten major dialects and hundreds of sub-dialects, some of which are markedly different from each other. Given the current situation of China's urban-rural dual economy and the massiveness of internal migration across dialect regions, the diversity of dialects affects the dissemination of information and technology, potentially hindering social and economic development [15]. Moreover, dialects are characterized by regional heterogeneity. Long-term usage of a specific dialect can reduce communication efficiency between speakers of different dialects, creating a "clustering" effect that can impact the social network and income levels of dialect speakers. Conversely, Standard Mandarin acts as a bridge of communication between speakers of different dialects and is an important symbol of social identity [18–19]. Therefore, language

proficiency is an important aspect of human capital for family entrepreneurs. Furthermore, the language used by family entrepreneurs affects their ability to integrate into different social networks, gain trust and support, and compensate for potential deficiencies in social capital through these networks [20].

The main objective of this paper is to investigate the formation of human capital and social capital among family entrepreneurs from the perspective of language and to examine its impact on family entrepreneurship choice and performance. Specifically, we examine the theoretical and empirical impact of Standard Mandarin and dialects on family entrepreneurship choice and performance, considering urban-rural differences. This study aims to enrich the current literature on the economics of language and provide a new perspective for future research on family entrepreneurship in both urban and rural areas, contributing to the understanding of how language proficiency influences entrepreneurial outcomes in the context of China's "mass entrepreneurship" and "street vending" policies. It also opens up new avenues for exploring how linguistic diversity and proficiency affect entrepreneurial outcomes in different cultural and economic contexts.

## Theoretical background and hypotheses development

Jacob Marschak, the founder of informatics, was the first to use the term "Economics of language", proposing that language is an indispensable element for economic activities and possesses the characteristics of economics, including value, utility, cost, and return [21].In the 1980s, researchers further integrated the theory of human capital and education economics into the economics of language, finding that language proficiency can be considered not only a marker of individual identity but also an investment in the form of human capital. Improved linguistic competence enables individuals to perform better in social situations, thereby obtaining more information and influencing their social and economic status [22]. For example, Di and Tansel found that mastering a foreign language has a positive effect on workers' income [23]. Azam et al. argued that mastering English has a premium effect on individual earnings in countries like India, where English is notthe official language, based on the 2005 data of the Anthropological Survey of India [24]. Gao and Smyth discovered that speaking standard Mandarin enhances the wage income of internal migrants, based on 2005 labor market data from 12 Chinese cities [25]. Thus, a good command of a widely spoken language in a country can be seen as a symbol of personal quality or an investment in the form of human capital [22].

In China, language is mainly categorized into standard Mandarin and regional dialects. The relationship between Standard Mandarin and dialects is somewhat analogous to that between English and other languages in other countries, with implications for the economics of language [25]. the As a form of human capital, standard Mandarin proficiency also affects the entrepreneurial behavior of families. Firstly, strong standard Mandarin reading and speaking skill can enhance the cognitive competence of family entrepreneurs. Alves and Yang argue that cognitive competence, which encompasses the ability to acquire and process information through memorization, comprehension, and expression, is a critical component of human capital [26]. Enhanced cognitive competence facilitates the acquisition and processing of information, enabling individuals to identify valuable information and acquire advanced entrepreneurial knowledge more efficiently. This, in turn, reduces the constraints on entrepreneurial inputs and assists entrepreneurs in making better decisions [27]. Secondly, given the complex entrepreneurial environment and information asymmetry, it can be challenging for transaction parties to accurately evaluate each other's capabilities to fulfill contractual obligations. Proficiency in Standard Mandarin can serve as a signal that helps family entrepreneurs demonstrate their abilities [28]. Entrepreneurs who are proficient in Standard Mandarin can effectively communicate their competencies in listening, speaking, reading, and writing, which are recognized as signals of high human capital. This can lead to a sense of identity and reduce transaction costs and uncertainty through clear and effective communication. Lastly, family entrepreneurs often rely not only on their resources but also on support from large financial institutions, where the primary language of communication is Standard Mandarin [29]. Entrepreneurs who speak Standard Mandarin can more easily demonstrate their competencies, gain recognition, and reduce the barriers to accessing entrepreneurial information and resources. This proficiency in Standard Mandarin not only enhances the cognitive ability of family entrepreneurs but also allows their personal qualities to be recognized,

thereby lowering the threshold for access to entrepreneurial information and resources and influencing entrepreneurship choice [28]. Based on the theoretical analysis above, we propose the following hypotheses:

**H1: The use of standard Mandarin significantly increases the probability of family entrepreneurship.**

**H2: A mediating effect of cognitive competence exists in the process of standard Mandarin influencing family entrepreneurship probability.**

Language also influences people's social networks, which in turn affects families' entrepreneurship behavior. In addition to large financial institutions, other financial sources are also important for families seeking monetary support when starting their own businesses, with relatives and friends playing a significant role [30]. In daily life, individuals in a social network may come from different regions, and standard Mandarin, as a common language, enables effective communication between individuals and increases the sense of mutual trust. bilingual individuals can also act as a bridge of communication between dialect-speaking friends and those who primarily use Standard Mandarin, occupying strategic positions in social networks [28]. Chen finds that standard Mandarin is a survival skill for internal migrant workers and acts as an important factor enabling them to be accepted by the local community [31]. Xu et al. demonstrate that the use of standard Mandarin can mitigate the negative impact of regional barriers on factor flow, formed by dialects, and improve the efficiency of resource allocation [15]. Wang et al. find that the improvement of standard Mandarin skills enables individuals to connect to more diverse groups and social strata, removing constraints on social networks imposed by kinship and geographic relationships [32]. Therefore, a strong skill in standard Mandarin is a critical medium for individuals who wish to integrate into various social networks. The use of standard Mandarin helps families to expand their social network and make up for the lack of entrepreneurial resources, thus influencing the entrepreneurship choice of families [20].

Accordingly, we expect that:

**H3: A mediating effect of social networks exists in the process of standard Mandarin influencing family entrepreneurship probability.**

Previous studies have demonstrated that the use of standard Mandarin or0 regional dialects affects entrepreneurial families' financial performance. Yueh argues that being bilingual can help entrepreneurs expand their social network and remove communication barriers, thereby improving entrepreneurial families' financial performance [33]. Gao and Smyth find that speaking standard Mandarin can reduce the discrimination faced by internal migrant dealers from local consumers, gaining customers' recognition and improving the efficiency of transactions [22]. Jin et al. examine that the stronger dialect oracy internal migrants display, the stronger social network they will establish in the local community, based on tests of mediating effects [34]. Yang et al. also find that strangers who speak the same dialect are more likely to trust each other. indicating that dialect can affect social trust and communication through identity recognition [14]. From this perspective, entrepreneurs who use dialects are more likely to integrate into local social networks, which in turn leads to better entrepreneurial financial performance. The impact of using standard Mandarin or dialects on financial performance depends on whether family entrepreneurs can integrate into the social circle of potential Mandarin-speaking or dialect-speaking customers and gain their recognition or trust.

Thus, we expect that:

**H4a: The use of standard Mandarin has a positive effect on entrepreneurial families' performance**

**H4b: The use of regional dialects has a positive effect on entrepreneurial families' performance**

The differences in the dual economic structure between urban and rural areas in China lead to distinct entrepreneurial environments for urban and rural families. The disparities in living and educational conditions result in gaps in Standard Mandarin proficiency and the breadth of social networks [5]. Urban areas typically enjoy a higher level of economic development, where

standard Mandarin is more widely spoken. Furthermore, urban families have easier access to formal credit financing and face less exclusion. In contrast, rural areas are relatively lower in economic level and have a lower popularization rate of Standard Mandarin. Consequently, rural families face more severe problems of information asymmetry regarding entrepreneurial resources and experience greater constraints on resource acquisition. Additionally, rural family businesses are characterized by low investment amounts and lower return rates, requiring more initial capital [35]. However, rural family entrepreneurs have a stronger social network based on kinship and geographic ties, which provide valuable tangible and intangible resources and support [36]. Language can influence entrepreneurial families' access to entrepreneurial resources through large financial institutions or social networks. Therefore, in the face of distinct entrepreneurial environments, standard Mandarin or regional dialects may have different influences on the entrepreneurship behavior of urban and rural families.

Therefore, we propose the following hypothesis:

**H5: The use of standard Mandarin or dialects influences on family entrepreneurship behavior differently in urban and rural areas.**

## Data and variables

### Data source

The data used in this study are drawn from the Chinese Family Panel Studies of 2018(CFPS2018). This is a nationwide tracking survey conducted by the Institute of Social Science Survey (ISSS) at Peking University, covering over 14,000 households across 25 provinces and municipalities directly under the central government. The dataset is structured into four hierarchical levels: community, family, adult, and children, providing comprehensive insights into changes in China's society, economy, population, education, and health. These data offer a reliable foundation for academic research on Chinese family behavior. For the purposes of this study, undefined or missing data in the dataset were excluded. Based on the availability of macro data, a total of 14,136 samples were selected for analysis.

### Variable specification and descriptive statistics

(1) Explained variable: Family entrepreneurial behavior

Family entrepreneurial behavior is divided into two aspects: family entrepreneurship choice (ent) and family entrepreneurial performance (opinc). Family entrepreneurship choice (ent) : This is a dummy variable that indicates self-employment in the non-agricultural sector or the establishment of a new enterprise, distinct from salaried employment [5,29]. Families that choose to start a business are assigned a value of 1, otherwise it is 0. Family entrepreneurial performance (opinc): To enhance comparability between urban and rural families, we follow previous empirical methodologies [13] and use annual operating income to represent family entrepreneurial performance. We also apply a logarithmic transformation to entrepreneurial performance to reduce heteroscedasticity.

(2) Explanatory variable: Familial daily language (lagu).

Familial daily language (lagu) is a dummy variable. A value of 1 is assigned if the family uses Standard Mandarin, and a value of 0 is assigned if a regional dialect is used.

(3) Mediating variables:

 a. Cognitive ability (cog)

We follow previous empirical methodologies [37–39] and take the logarithmic treatment of the sum of scores on literacy, numerical ability, and memory derived from the CFPS2018 to quantify the cognitive competence of respondents. we then standardize these data.

b. Social network (social)

Chinese social networks are often rooted in Confucian culture, where connections are primarily based on kinship and geographical proximity [13]. Two predominant ways of maintaining relationships between relatives and friends are gift-giving during traditional festivals or events and daily greetings. We measure social networks by taking the logarithmic transformation of the sum of gift money spent by a family during festivals or events and communication expenses.(4) Control variables:

We control variables related to family entrepreneurship at three levels: individual characteristics, family particulars, and macroeconomy factors.

In terms of individual characteristics, we control householders' gender (gender), age(age), years of education (edu), life satisfaction(sta), future expectations(future), health condition(heth), risk appetite(prisk), trust(trust), social security status(sin), etc [9].

At the family particulars, we control the place of household registration(hk), household-saving(dep), family size(fsize), bank loan status (finbank) [19].

The regional level of the economy also affects family entrepreneurial behavior and is measured by the logarithm of per capita GDP of the province (gdp)where families are located [38]. The main variables are defined in Table 1.

Table 2 presents the descriptive statistics of the variables. As can be seen from Table 2, the proportion of entrepreneurial families (ent) in the nationwide sample is 12.80%, with urban entrepreneurial families constituting 16%, significantly higher than the 9.5% in rural entrepreneurial families. The mean logarithm of annual operating income (opinc) in the national sample is 3.646, with the corresponding value for urban families being 2.728, lower than that of rural families, which is 4.599. Approximately 57.70% of families use Standard Mandarin (lagu) in the nationwide sample, with 67.10% of urban families using Standard Mandarin in daily life, significantly higher than the 47.90% usage rate in rural families. Urban families score higher than rural families in social network (social), cognitive competence (cog), family savings (dep), household head's years of education (edu), health condition (heth), and regional per capita GDP (gdp). In terms of family size (fsize), bank loan status (finbank), household head's gender and age (gender, age), life satisfaction (sta), confidence in the future (future), social security status (sin), and confidence in society (trust), the values of urban families are generally lower than those of rural families. The value of risk appetite (prisk) is similar between urban and rural families.

## Models and empirical results

### Baseline model

A Probit model is used to estimate the influence of standard Mandarin and dialect on family entrepreneurship choice. The model is specified as follows:

$$ent_i^* = \alpha_1 + \beta_1 lagu_i + \delta control_i + \varepsilon_i$$
$$ent_i = 1(ent_i^* \succ 0)$$

(1)

Here, the error term $\varepsilon_i$ follows a normal distribution, $ent_i$ is a latent variable, with a value of 1 when a family chooses to start a business, and 0 otherwise. $lagu_i$ represents familial daily language, with a value of 1 if the family uses Standard Mandarin and 0 if they use a regional dialect.$control_i$ represents the control variable, containing characteristics of families, individuals, and macroeconomy.

When studying family entrepreneurial performance, only the operating income of families who started their own businesses can be observed. For families that didn't start their businesses, the observed value was 0, which can be seen as censored data. Therefore, a Tobit equation is used for estimation, with the model specified as follows:

$$y_i^* = \alpha_2 + \beta_2 lagu_i + \gamma control_i + \varepsilon_i$$
$$Y_i = (0, y_i^*)$$

(2)

**Table 1. Variable definition.**

| Variables | Abbreviation | Definitions |
|---|---|---|
| Family entrepreneurship choice | ent | If the family chooses to start a business, the value assigned is 1, otherwise it is 0. |
| Family entrepreneurial performance | opinc | Ln (Annual operating income +1) |
| Familial daily language | lagu | If the familial daily language used is standard Mandarin, the value assigned is 1, otherwise it is 0. |
| Cognitive competence | cog | Ln(Sum of householder's scores on literacy, numerical ability, and memory +1) , a higher value indicates stronger cognitive competence. |
| Social network | social | Ln(Annual household gift expenses + annual telephone charge + 1),a higher value indicates a stronger social network. |
| Householder's gender | gender | If the householder's gender is male, the value assigned is 1, otherwise it is 0. |
| Householder's age | age | Actual age |
| Householder's year of education | edu | Householder's education background, ranging from illiterate to PhD, the value assigned is 0–9. |
| Life satisfaction | sta | Rate of life satisfaction, measured from 1 to 5, a higher value indicates a higher satisfaction. |
| Confidence in the future | future | Rate of confidence in the future, measured from 1 to 5, a higher value indicates a higher confidence. |
| Health condition | heth | Self-rated health condition: 1–5 were assigned from 'very healthy' to 'unhealthy'. |
| Risk appetite | prisk | A risk-seeker is assigned a value of 1, while a risk-averter will be assigned a value of 0. |
| Confidence in society | trust | The respondents' confidence in society was measured by assigning values from 1 to 10, with the higher the confidence level, the greater the value. |
| Social security status | sin | If the respondent lacks a social security plan, the value assigned is 1, otherwise it is 0. |
| Family location | hk | According to classification by the National Bureau of Statistics, if a family is located in the urban area, the value assigned is 1, otherwise, it is 0. |
| Family savings | dep | Ln (current family savings balance+1) |
| Family size | fsize | Number of family members |
| Bank loan status | finbank | If a family borrows from a bank, the value assigned is 1, otherwise is 0. |
| Regional Per capita GDP | gdp | Ln (Per capita GDP of the province where a family is located) |

Here, $y_i^*$ is a latent variable representing the optimal family entrepreneurial performance; $Y_i$ represents the observable family entrepreneurial performance, measured by the logarithm of annual operating income. $control_i$ is the vector of control variables.

## Empirical results and analysis

(1) The influence of language on family entrepreneurial behavior

Table 3 presents the empirical results of the influence of standard Mandarin and dialects on family entrepreneurship choice and entrepreneurial performance based on a nationwide sample. Equations (1) and (3) are based on OLS linear regression results, Equation (2) is based on Probit regression results, and Equation (4) is based on Tobit regression results. According to Equations (1) and (2), when controlling for variables at the individual, family, and regional economy levels, familial daily language (lagu) has a positive influence on family entrepreneurship choice (ent), which is significant at the 99% confidence level in both OLS and Probit regression. The marginal effect of standard Mandarin on family entrepreneurship choice is approximately 0.0192, indicating that families using standard Mandarin in daily life are more likely to start their businesses. This supports Hypothesis H1. This finding may be because the use of standard Mandarin can help families gain advantages in social and human capital, and access to more entrepreneurial resources, thus influencing family entrepreneurial behaviors [28,40].

**Table 2. Descriptive statistics.**

| Variables | Nationwide sample | | | Urban sample | | | Rural sample | | |
|---|---|---|---|---|---|---|---|---|---|
| | Mean | Std.dev. | Obs. | Mean | Std.dev. | Obs. | Mean | Std.dev. | Obs. |
| Family entrepreneurship choice | 0.128 | 0.334 | 14136 | 0.160 | 0.366 | 7203 | 0.095 | 0.293 | 6933 |
| Family entrepreneurial performance | 3.646 | 4.509 | 14136 | 2.728 | 4.394 | 7203 | 4.599 | 4.429 | 6933 |
| Familial daily language | 0.577 | 0.494 | 14136 | 0.671 | 0.470 | 7203 | 0.479 | 0.500 | 6933 |
| Social network | 2.792 | 0.103 | 14136 | 2.806 | 0.101 | 7203 | 2.780 | 0.104 | 6933 |
| Cognitive competence | 3.353 | 0.817 | 14136 | 3.546 | 0.639 | 7203 | 3.173 | 0.919 | 6933 |
| Family size | 4.477 | 2.04 | 14136 | 4.111 | 1.932 | 7203 | 4.858 | 2.080 | 6933 |
| Bank loan status | 0.123 | 0.328 | 14136 | 0.110 | 0.313 | 7203 | 0.136 | 0.343 | 6933 |
| Family savings | 7.436 | 4.583 | 14136 | 8.142 | 4.500 | 7203 | 6.703 | 4.554 | 6933 |
| Householder's gender | 0.496 | 0.500 | 14136 | 0.488 | 0.500 | 7203 | 0.504 | 0.500 | 6933 |
| Householder's age | 35.473 | 13.55 | 14136 | 34.79 | 12.838 | 7203 | 36.183 | 14.218 | 6933 |
| Householder's year of education | 3.892 | 2.022 | 14136 | 4.490 | 1.873 | 7203 | 3.270 | 1.984 | 6933 |
| Health condition | 2.742 | 1.130 | 14136 | 2.753 | 1.065 | 7203 | 2.732 | 1.194 | 6933 |
| Life satisfaction | 3.945 | 0.931 | 14136 | 3.922 | 0.896 | 7203 | 3.969 | 0.965 | 6933 |
| Confidence in the future | 4.171 | 0.878 | 14136 | 4.136 | 0.852 | 7203 | 4.207 | 0.902 | 6933 |
| Social security status | 0.894 | 0.308 | 14136 | 0.883 | 0.322 | 7203 | 0.905 | 0.293 | 6933 |
| Confidence in society | 6.703 | 1.952 | 14136 | 6.625 | 1.894 | 7203 | 6.785 | 2.006 | 6933 |
| Risk appetite | 0.712 | 0.453 | 14136 | 0.712 | 0.453 | 7203 | 0.713 | 0.453 | 6933 |
| Regional Per capita GDP | 10.904 | 0.395 | 14136 | 11.007 | 0.412 | 7203 | 10.797 | 0.346 | 6933 |

From Equations (3) and (4), it can be seen that the marginal effect of Mandarin on family entrepreneurial performance is about -0.6300 after controlling variables of three levels, and is significant at the 99% confidence level in both OLS and Tobit regression. This supports hypothesis H4b and rejects H4a. It shows that the entrepreneurial performance of families using dialects is higher than that of families using standard Mandarin as their daily language. This may be because many families' entrepreneurial businesses, often referred to as 'street vendors', face customers who are mostly local people in the community. The use of dialect helps the entrepreneurs to gain trust and more easily integrate into the local community, thus achieving better entrepreneurial performance [14,34].

Among the family control variables, family size (fsize) has a positive impact on family entrepreneurship choice and performance, with both impacts being significant at the 99% confidence level. This is because a larger family size provides more labor force for family entrepreneurship [41]. Bank loan status (finbank) has a significant positive influence on family entrepreneurship choice, aligning with the conclusions of Weng and Zhang [29]. A positive bank loan status suggests that a family faces less financial exclusion and is capable of financing through large financial institutions, thereby influencing entrepreneurship choice Family savings (DEP) has a positive impact on family entrepreneurship choice and performance. Family savings represent the level of family wealth, and the higher the level of family wealth, the more likely a family is to choose to start a business and the more focus it can put to family business activities [29].

In the control variables of the individual (the householder), the gender of the householder has no significant impact on entrepreneurship choice but has a significant positive impact on entrepreneurial performance, indicating that males have more advantages in entrepreneurial activities. The age of the householder has no significant influence on the entrepreneurship choice but has a negative influence on entrepreneurial performance, significant at the 95% confidence level, indicating that younger household heads are associated with higher family business income.. This may be linked to the younger householders' vigor and learning capacity. The householder's years of education (edu) has a significant positive impact on entrepreneurship choice, indicating a trend toward higher-educated individuals engaging in family entrepreneurship in recent years, although this differs from the conclusion of Yin, Gong, and Guo [41]. However, the impact of years of

**Table 3. The influence of language on family entrepreneurial behavior.**

| Variables | Abbreviations | ent (1) ols | ent (2) probit | opinc (3) ols | opinc (4) tobit |
|---|---|---|---|---|---|
| Familial daily language | lagu | 0.0192***(0.0061) | 0.0198***(0.0062) | -0.6300***(0.0705) | -0.6300***(0.0675) |
| Family size | fsize | 0.0127***(0.0014) | 0.0124***(0.0013) | 0.3158***(0.0161) | 0.3158***(0.0159) |
| Bank loan status | finbank | 0.0701***(0.0098) | 0.0623***(0.0079) | -0.0877(0.0971) | -0.0877(0.0952) |
| Family savings | dep | 0.0045***(0.0006) | 0.0045***(0.0007) | 0.0334***(0.0070) | 0.0334***(0.0070) |
| Householder's gender | gender | -0.0039(0.0056) | -0.0041(0.0056) | 0.1056*(0.0623) | 0.1056*(0.0624) |
| Householder's age | age | 0.0005*(0.0002) | 0.0004(0.0003) | -0.0103***(0.0029) | -0.0103***(0.0029) |
| Householder's year of education | edu | 0.0091***(0.0016) | 0.0097***(0.0017) | -0.2760***(0.0194) | -0.2760***(0.0196) |
| Health condition | heth | -0.0017(0.0026) | -0.0019(0.0027) | -0.0886***(0.0304) | -0.0886***(0.0297) |
| Life satisfaction | sta | -0.0011(0.0029) | -0.0013(0.0030) | 0.0051(0.0389) | 0.0051(0.0379) |
| Social security status | sin | 0.0093(0.0088) | 0.0098(0.0094) | 0.3916***(0.0952) | 0.3916***(0.1014) |
| Risk appetite | prisk | -0.0089(0.0063) | -0.0093(0.0062) | -0.0423(0.0690) | -0.0423(0.0690) |
| Regional per capital GDP | gdp | 0.0087(0.0073) | 0.0088(0.0072) | -1.3572***(0.0751) | -1.3572***(0.0832) |
| Family entrepreneurship choice | ent | | | 6.9023***(0.0883) | 6.9023***(0.0930) |
| Confidence in the society | trust | | | 0.0429***(0.0164) | 0.0429***(0.0162) |
| Confidence in the future | future | | | 0.053(0.0410) | 0.053(0.0399) |
| | N | 14,136 | 14,136 | 14,136 | 14,136 |
| | adj. R² | 0.0177 | | 0.3412 | |

Notes: standard errors in parentheses, and P-value of the corresponding tests in square brackets.

*Denotes p < 0.10.

**Denotes p < 0.05.

***Denotes p < 0.01.

education on family entrepreneurial performance is significantly negative, indicating that people with higher education still need to integrate theory with practice to achieve better entrepreneurial performance. Both social security status (sin) and confidence in society (trust) can promote family entrepreneurial performance.

In regional economy variables, the regional per capita GDP (gdp) has a negative influence on family entrepreneurial performance significant at the 99% confidence level. This may be because family businesses are typically small in size and more susceptible to competition in economically developed areas. Therefore, the higher the level of regional economy, the worse the performance of family entrepreneurship.

(2) Analysis of rural-urban differences

Table 4 presents the empirical results of the impact of Mandarin and dialect on family entrepreneurship choice based on urban and rural sub-samples. Equations (5) and (6) are OLS and Probit regression results based on urban samples, and Equations (7) and (8) are the results of rural samples. According to the empirical results, the daily use of Mandarin has no significant impact on the entrepreneurship choice of rural families, but it has a positive impact on urban families, which is significant at the 99% confidence level. This indicats that the effect of Standard Mandarin on the entrepreneurship choice of families is more obvious in urban areas, supporting hypothesis H5. This is likely due to the higher popularizing rate of Standard Mandarin in urban areas (67.10%), enabling Mandarin-speaking urban families to better access the resources required to start a business through social networks and other means. In rural areas, the popularizing rate of Mandarin (47.90%) is similar to that of dialect (52.10%). In terms of social network and communication efficiency, Mandarin-speaking families do not have a significant advantage over those using dialect. Therefore, the influence of using Standard

**Table 4. Effects of familial daily language on entrepreneurship choice: based on sub-samples.**

| Variables | Abbreviations | Urban sample | | Rural sample | |
|---|---|---|---|---|---|
| | | (5) | (6) | (7) | (8) |
| | | ols | probit | ols | probit |
| Familial daily language | lagu | 0.0361***(0.0099) | 0.0358***(0.0099) | -0.001(0.0073) | -0.001(0.0073) |
| Family size | fsize | 0.0255***(0.0024) | 0.0238***(0.0021) | 0.0043***(0.0016) | 0.0042***(0.0015) |
| Bank loan status | finbank | 0.0908***(0.0157) | 0.0809***(0.0126) | 0.0628***(0.0120) | 0.0536***(0.0093) |
| Family savings | dep | 0.0046***(0.0010) | 0.0046***(0.0010) | 0.0039***(0.0008) | 0.0040***(0.0008) |
| Householder's gender | gender | -0.001(0.0086) | -0.0016(0.0086) | -0.0027(0.0071) | -0.0029(0.0070) |
| Householder's age | age | -0.0002(0.0004) | -0.0002(0.0004) | 0.0005*(0.0003) | 0.0005(0.0003) |
| Householder's year of education | edu | -0.0057**(0.0026) | -0.0055**(0.0026) | 0.0143***(0.0021) | 0.0152***(0.0023) |
| Health condition | heth | -0.0027(0.0043) | -0.0027(0.0042) | -0.0013(0.0030) | -0.0015(0.0032) |
| Life satisfaction | sta | -0.0026(0.0049) | -0.0023(0.0049) | 0.0002(0.0033) | -0.0001(0.0036) |
| Social security status | sin | 0.012(0.0130) | 0.012(0.0140) | 0.0126(0.0113) | 0.0132(0.0123) |
| Risk appetite | prisk | -0.0123(0.0097) | -0.0132(0.0095) | -0.0095(0.0080) | -0.0085(0.0076) |
| Regional Per capita GDP | gdp | -0.0390***(0.0100) | -0.0404***(0.0105) | 0.0492***(0.0114) | 0.0463***(0.0102) |
| | N | 7,203 | 7,203 | 6,933 | 6,933 |
| | adj. R | 0.0286 | | 0.0227 | |

Notes: standard errors in parentheses, and P-value of the corresponding tests in square brackets.

*Denotes $p < 0.10$.

**Denotes $p < 0.05$.

***Denotes $p < 0.01$.

Mandarin in rural families on access to entrepreneurial resources is not significant. In addition, we find that the impact of the householder's years of education on the entrepreneurship choice of urban and rural families differs. In urban areas, the more years of education a household head has, the lower the probability that the family will choose to start a business. Conversely, in rural areas, the opposite is true. This may be because in urban areas, with higher levels of economic development, individuals with higher education have a higher probability of securing decent positions, which inhibits their choice to start a business. However, in rural areas, there are fewer opportunities to find decent jobs, individuals with higher education are more inclined to start a business instead.

Table 5 presents the empirical results of the impact of Standard Mandarin and dialect on family entrepreneurial performance based on urban and rural sub-samples. Equations (9) and (11) are OLS regressions and Equations (10) and (11) are Tobit models. The daily uses of Standard Mandarin by both urban and rural families has a negative impact on entrepreneurial performance, which is significant at 99% confidence level. This indicates that both urban and rural families using dialect are more likely to have better entrepreneurial performance. This may be because, regardless of whether in urban or rural areas, family entrepreneurial businesses tend to be small-scale and most of the customers are local residents within the community. Dialects can help family entrepreneurs more easily gain trust and integrate into the local community [13,34], thereby achieving better entrepreneurial performance.

(3) Discussion of the underlying mechanism

In this part, we explore the underlying mechanisms through which language affects families' entrepreneurial behavior.

We suppose two potential mechanisms. Firstly, familial daily language can affect family entrepreneurs 'cognitive competence. Proficiency in Standard Mandarin not only enhances the effectiveness of processing complex commercial information and acquiring advanced business knowledge but also serves as a signal of personal quality to business partners, reducing the cost of transaction and information asymmetry in entrepreneurship. Therefore, the use of standard

**Table 5. Effects of familial daily language on entrepreneurial performance: based on sub-samples.**

| Variables | Abbreviations | Urban sample | | Rural sample | |
|---|---|---|---|---|---|
| | | (9) | (10) | (11) | (12) |
| | | ols | tobit | ols | tobit |
| Familial daily language | lagu | -0.7828***(0.0887) | -0.7828***(0.0804) | -0.3757***(0.1033) | -0.3757***(0.1029) |
| Family size | fsize | 0.2259***(0.0197) | 0.2259***(0.0192) | 0.2895***(0.0236) | 0.2895***(0.0241) |
| Bank loan status | finbank | -0.1472(0.1142) | -0.1472(0.1135) | -0.102(0.1455) | -0.102(0.1449) |
| Family savings | dep | 0.0505***(0.0080) | 0.0505***(0.0081) | 0.0453***(0.0110) | 0.0453***(0.0109) |
| Householder's gender | gender | 0.0719(0.0709) | 0.0719(0.0709) | 0.0109(0.0993) | 0.0109(0.0993) |
| Householder's age | age | -0.0073**(0.0033) | -0.0073**(0.0034) | 0.0004(0.0045) | 0.0004(0.0045) |
| Householder's year of education | edu | -0.2145***(0.0233) | -0.2145***(0.0232) | -0.0603*(0.0321) | -0.0603*(0.0319) |
| Health condition | heth | -0.0804**(0.0376) | -0.0804**(0.0357) | -0.065(0.0446) | -0.065(0.0446) |
| Life satisfaction | sta | -0.0249(0.0468) | -0.0249(0.0452) | 0.0658(0.0578) | 0.0658(0.0575) |
| Social security status | sin | 0.2575***(0.0987) | 0.2575**(0.1107) | 0.4686***(0.1658) | 0.4686***(0.1687) |
| Risk appetite | prisk | -0.089(0.0795) | -0.089(0.0787) | 0.0788(0.1089) | 0.0788(0.1092) |
| Regional Per capita GDP | gdp | -0.8776***(0.0800) | -0.8776***(0.0887) | -1.0635***(0.1428) | -1.0635***(0.1481) |
| Family entrepreneurship choice | ent | 8.2230***(0.1065) | 8.2230***(0.0969) | 5.2976***(0.1422) | 5.2976***(0.1684) |
| Confidence in society | trust | 0.0343*(0.0193) | 0.0343*(0.0190) | 0.0173(0.0250) | 0.0173(0.0251) |
| Confidence in the future | future | -0.0281(0.0487) | -0.0281(0.0472) | 0.0815(0.0618) | 0.0815(0.0610) |
| | N | 7,203 | 7,203 | 6,933 | 6,933 |
| | adj. R | 0.5438 | | 0.1603 | |

Notes: standard errors in parentheses, and P-value of the corresponding tests in square brackets.

*Denotes p<0.10.

**Denotes p<0.05.

***Denotes p<0.01.

Mandarin helps to improve the cognitive competence of entrepreneurs, which enables them to access more entrepreneurial resources, thereby influencing the entrepreneurship choice of families [28]. Secondly, familial daily language can affect family entrepreneurs ' social networks. a strong skill in standard Mandarin is a critical medium for individuals who wish to integrate into various social networks. The use of standard Mandarin helps families to expand their social network and make up for the lack of entrepreneurial resources, thus influencing entrepreneurial behaviors.Based on the above analysis, we use the logarithm of the cognitive competence of the householder (cog) to measure the formation of family cognitive ability [37–39]. Given the deep influence of Confucian culture in China, family social networks(social) are primarily based on kinship and geographic ties [13]. Therefore, we measure the family social network by taking the logarithm of the sum of gift money and family communication costs that a family actively spends on relatives and friends during traditional festivals and major events [34].

This study employs the Karlson, Holm, and Breen (KHB) mediation analysis method to decompose and statistically assess the pathways through which household daily language influences family entrepreneurial behavior. Expanding on a baseline model, we introduced two mediating variables—social network (social) and cognitive ability(cog)—to determine their respective contributions and identify which mediator exerts a stronger indirect effect.

The findings, summarized in Table 6, indicate that both the total effect and the direct effect of household daily language on family entrepreneurial behavior are positive and statistically significant. This suggests that the language used within a household context has a direct, beneficial impact on the entrepreneurial activities of its members.

Regarding the indirect effects, both social network and cognitive ability were found to significantly mediate the relationship between household daily language and family entrepreneurial behavior. Social network accounted for 67.98% of

**Table 6. ï The mediating effect of familial daily language(lagu) on family entrepreneurship choice(ent).**

| Variables | social | cog |
|---|---|---|
| Reduced | 0.149*** ( 0.033 ) | 0.143*** ( 0.033 ) |
| Full | 0.059* ( 0.034 ) | 0.087*** ( 0.033 ) |
| Diff | 0.091*** ( 0.008 ) | 0.056*** ( 0.008 ) |
| Components of Diff | 67.98% | 32.02% |
| Components of Full | 55.64% | 26.20% |
| N | 9192 | 9192 |

Notes: standard errors in parentheses, and P-value of the corresponding tests in square brackets.

*Denotes p<0.10.

**Denotes p<0.05.

***Denotes p<0.01.

the indirect effect, while cognitive ability contributed 32.02%. Collectively, these mediators explained 81.84% of the total effect, underscoring their critical role in this relationship.

These results support Hypotheses H2 and H3, which posited that both social network and cognitive ability would significantly mediate the impact of household daily language on family entrepreneurial behavior. The dominance of social network as a mediator implies that the extent and quality of a family's social connections play a pivotal role in translating household linguistic practices into entrepreneurial outcomes.

(4) Tests on endogeneity

In the baseline model, the influence of familial daily language (Standard Mandarin or dialect) on the family entrepreneurial behavior may be subject to bidirectional causality and omission variables, leading to endogeneity issues. Specifically, On the one hand, the daily use of standard Mandarin or dialect in a family can affect entrepreneurship choice and performance. To improve the efficiency of communication with providers of entrepreneurial resources or customers during the operation of the family business, entrepreneurial families may change their habits of using standard Mandarin or dialect. On the other hand, the use of standard Mandarin or dialect in families may be influenced by local customs, attitudes toward accepting new things, and other factors that are not observable in our dataset.

To address the potential endogeneity issues discussed earlier, this paper employs the instrumental variable (IV) method. The formation of a habit of using Standard Mandarin or dialect as a familial daily language is a long process that can be influenced by past habits. Current entrepreneurial behavior, however, is unlikely to have a direct impact on the previous habit of using Standard Mandarin or dialect in a family. Therefore, the familial daily language (lagu) from 2016 is selected as the instrumental variable of the explanatory variable (lagu), so as to conduct a two-stage instrumental variable estimation.Table 7 shows the results of the endogeneity test. From the test results , It can be seen that the F-values of Equations (13) to (18) in the first stage are all greater than the critical value of 16.38 at the 10% error level [41–42], leading us to reject the hypothesis of a weak instrumental variable. The Wald tests for Equations (13) to (15) yield results of 0.63, 1.05, and 0.36, respectively, none of which are significant and do not pass the test. This indicates that there is no evidence of endogeneity in the explanatory variables regarding the influence of Standard Mandarin or dialect on family entrepreneurship choice. The Wald test results for Equations (16) to (18) are 18.78, 2.73, and 8.57, respectively, all of which are significant at least 90% confidence levels All of them pass the Wald test, indicating that the influence of standard Mandarin or dialect on family entrepreneurial performance is endogenetic. The regression results using instrumental variables in Equations (16) to (18) show that, when the effect

 

**Table 7. Endogeneity test: Instrument variables.**

| Variables | Abbreviations | Nationwide | Urban | Rural | Nationwide | Urban | Rural |
|---|---|---|---|---|---|---|---|
| | | (13) | (14) | (15) | (16) | (17) | (18) |
| | | probit | probit | probit | tobit | tobit | tobit |
| Familial daily language | iv_lagu | 0.0132*(0.0080) | 0.0213*(0.0122) | -0.0035(0.0098) | -0.5856***(0.0891) | -0.4526***(0.1009) | -0.5081***(0.1431) |
| Family size | fsize | 0.0142***(0.0018) | 0.0243***(0.0029) | 0.0070***(0.0022) | 0.3282***(0.0229) | 0.2457***(0.0269) | 0.3053***(0.0361) |
| Bank loan status | finbank | 0.0627***(0.0106) | 0.0807***(0.0165) | 0.0528***(0.0129) | -0.079(0.1326) | -0.0366(0.1516) | -0.1469(0.2103) |
| Family savings | dep | 0.0046***(0.0009) | 0.0049***(0.0014) | 0.0037***(0.0011) | 0.0219**(0.0094) | 0.0296***(0.0106) | 0.0436***(0.0151) |
| Householder's gender | gender | 0.0016(0.0075) | 0.0046(0.0112) | 0.0016(0.0096) | 0.1029(0.0844) | 0.0544(0.0932) | -0.0192(0.1389) |
| Householder's age | age | 0.0003(0.0004) | -0.0006(0.0005) | 0.0007(0.0004) | -0.0130***(0.0040) | -0.0108**(0.0045) | 0.0001(0.0064) |
| Householder's year of education | edu | 0.0084***(0.0023) | -0.0056(0.0034) | 0.0143***(0.0032) | -0.3216***(0.0263) | -0.2473***(0.0301) | -0.1088**(0.0447) |
| Health condition | heth | -0.0072**(0.0036) | -0.0099*(0.0057) | -0.0053(0.0043) | -0.0902**(0.0402) | -0.0816*(0.0473) | -0.0717(0.0619) |
| Life satisfaction | sta | -0.0044(0.0040) | -0.0056(0.0065) | -0.0045(0.0047) | -0.0038(0.0511) | -0.0011(0.0595) | 0.0534(0.0795) |
| Social security status | sin | 0.0012(0.0120) | 0.0045(0.0173) | 0.0052(0.0165) | 0.4805***(0.1336) | 0.2346*(0.1401) | 0.6127***(0.2342) |
| Risk appetite | prisk | -0.0046(0.0082) | -0.0136(0.0123) | 0.0042(0.0106) | -0.0243(0.0937) | -0.0826(0.1027) | 0.1134(0.1547) |
| Regional per capital GDP | gdp | 0.007(0.0095) | -0.0343**(0.0134) | 0.0384***(0.0137) | -1.3085***(0.1111) | -0.8668***(0.1149) | -1.0017***(0.2062) |
| Family entrepreneurship choice | ent | | | | 6.9809***(0.1303) | 8.3107***(0.1323) | 5.2242***(0.2438) |
| Confidence in the society | trust | | | | 0.0490**(0.0219) | 0.0451*(0.0250) | 0.0295(0.0348) |
| Confidence in the future | future | | | | 0.022(0.0538) | -0.0675(0.0620) | 0.0719(0.0842) |
| | | 7,445 | 3,921 | 3,524 | 7,445 | 3,921 | 3,524 |
| | | 209.23*** | 116.56*** | 95.86*** | 192.48*** | 93.58*** | 77.32*** |
| | | 0.63 | 1.05 | 0.36 | 18.78*** | 2.73* | 8.57*** |

Notes: standard errors in parentheses, and P-value of the corresponding tests in square brackets.

*Denotes p < 0.10.

**Denotes p < 0.05.

***Denotes p < 0.01.

of endogeneity is excluded, the entrepreneurial performance of families using Standard Mandarin is poorer, while the performance of families using dialect is better. These findings are consistent with the regression results presented earlier, suggesting that there is no substantial difference in the conclusions drawn from the original and instrumental variable models.

(5) Tests on heterogeneity

Considering the respondents' individual traits, this paper focuses on the heterogeneity in how the family income, the householder's age, and the dialect area of the family are affected differently by Standard Mandarin and dialect. This analysis helps understanding the impact of standard Mandarin and dialects on families' entrepreneurship choices and for the government to devise targeted incentive policy for family entrepreneurship.

According to the Language Atlas of China (1990), China is divided into ten major dialect regions: Northern Mandarin, Jin, Wu, Hui, Gan, Hunan, Min, Cantonese, Pinghua, and Hakka. Within the Northern Mandarin region, there are further divisions, including Northeast Mandarin, Beijing Mandarin, Yilu Mandarin, Jiao-Liao Mandarin, Central Mandarin, Lanyin Mandarin, Jianghuai Mandarin, and Southwest Mandarin. There are significant differences not only between dialect areas but also between dialects within different Mandarin areas, with clear communication barriers among these language areas.

Table 8 shows the empirical results based on the influence of familial daily language (lagu) on family entrepreneurship choice (ent) and entrepreneurial performance (opinc) in the dialect areas sample. As can be seen from the regression results, familial daily languages (lagu) have significantly positive impacts on family entrepreneurship choice (ent) in most regional groups, while there are significantly negative impacts on family entrepreneurial performance (opinc). This is consistent with the baseline regression results.. In Hunan, Gan, and Pinghua, the impact of familial daily language (lagu) on the family entrepreneurship choice (ent) is not significant, indicating that the use of standard Mandarin does not help families gain more entrepreneurial resources or information in these areas, thus not affecting their entrepreneurship choice. In Min, Hunan and Pinghua, the impact of familial daily language (lagu) on entrepreneurial performance (opinc) is not significant, suggesting that the use of dialect does not help families integrate into the local community, with the business performance of the family hardly influenced.

Table 9 presents the empirical results of heterogeneity analysis by income, age, and region. According to the regression results, in high-income, young, eastern families, and central families, daily language (lagu) has a significant positive impact on entrepreneurship choice (ent), This indicates that families using Standard Mandarin have a higher probability of starting a business.. However, for low-income, middle-aged, and western families, the influence of daily language (lagu) on their entrepreneurship choice is not significant.

**Table 8. The influence of familial daily language on entrepreneurial behavior: regional sub-sample analysis.**

| Dialect areas | Variables | Choice | Performance | Dialect areas | Variables | Choice | Performance |
|---|---|---|---|---|---|---|---|
| Northern Mandarin | Lagu | 0.0125*(0.0069) | -0.6980*** (0.0765) | Cantonese | lagu | 0.0563***(0.0187) | -0.3067*(0.1816) |
| | Control variables | Control | Control | | Control variables | Control | Control |
| | N | 11,473 | 11,473 | | N | 1,593 | 1,593 |
| Jin | Lagu | 0.0271**(0.0122) | -0.9206***(0.1370) | Hunan | lagu | 0.0169(0.0250) | -0.208(0.3109) |
| | Control variables | Control | Control | | Control variables | Control | Control |
| | N | 3,339 | 3,339 | | N | 1,280 | 1,280 |
| Wu | Lagu | 0.0901***(0.0256) | -0.7601***(0.1932) | Gan | lagu | 0.0281(0.0351) | -0.9996***(0.2148) |
| | Control variables | Control | Control | | Control variables | Control | Control |
| | | 1,913 | 1,913 | | N | 689 | 689 |
| Min | Lagu | 0.0833***(0.0178) | -0.1241(0.1544) | Hui | lagu | 0.1236***(0.0443) | -1.0512***(0.3281) |
| | Control variables | Control | Control | | Control variables | Control | Control |
| | N | 2,070 | 2,070 | | N | 879 | 879 |
| Hakka | Lagu | 0.0508***(0.0183) | -0.3625**(0.1587) | Pinghua | lagu | 0.0281(0.0351) | -0.208(0.3109) |
| | Control variables | Control | Control | | Control variables | Control | Control |
| | N | 2,076 | 2,076 | | N | 689 | 689 |

Notes: standard errors in parentheses, and P-value of the corresponding tests in square brackets.

*Denotes p < 0.10.

**Denotes p < 0.05.

***Denotes p < 0.01.

 

**Table 9. Heterogeneity test by income, age, and region.**

| Groups | | variables | ent | opinc |
|---|---|---|---|---|
| Grouped by income | High-income | lagu | 0.0220**(0.0101) | -0.6395***(0.0860) |
| | | Control variables | Control | Control |
| | | N | 7,286 | 7,286 |
| | Low-income | lagu | 0.002(0.0067) | -0.6314***(0.1063) |
| | | Control variables | Control | Control |
| | | N | 6,850 | 6,850 |
| Grouped by age | Old | lagu | 0.0027(0.0117) | -0.5744***(0.1654) |
| | | Control variables | Control | Control |
| | | N | 2,333 | 2,333 |
| | Young | lagu | 0.0232***(0.0070) | -0.6107***(0.0740) |
| | | Control variables | Control | Control |
| | | N | 11,803 | 11,803 |
| Grouped by area | Eastern | lagu | 0.0192*(0.0110) | -0.1333(0.1060) |
| | | Control variables | Control | Control |
| | | N | 5,417 | 5,417 |
| | Central | lagu | 0.0441***(0.0117) | -1.1720***(0.1227) |
| | | Control variables | Control | Control |
| | | N | 4,172 | 4,172 |
| | Western | lagu | -0.0083(0.0102) | -0.8745***(0.1311) |
| | | Control variables | Control | Control |
| | | N | 4,547 | 4,547 |

Notes: standard errors in parentheses, and P-value of the corresponding tests in square brackets.

*Denotes $p < 0.10$.

**Denotes $p < 0.05$.

***Denotes $p < 0.01$.

Furthermore, familial daily language has a significant negative impact on entrepreneurial performance (opinc) in most of the regional sub-samples, suggesting that the use of regional dialects helps to improve entrepreneurial performance.. Notably, in the eastern region, the influence of familial daily language (lagu) on family entrepreneurship choice is not significant. This may be due to the high popularization rate of Standard Mandarin in the economically developed eastern region (73.21%), which renders the marginal effect of using Standard Mandarin or dialects less pronounced.

## Discussion

Our results demonstrate that families who predominantly use Standard Mandarin have a higher probability of starting a business, with cognitive competence and social networks serving as two important channels through which language influences the likelihood of entrepreneurship. Specifically, we provide evidence that the effect of Standard Mandarin on urban families' entrepreneurship choice is more significant, whereas the impact on rural families is not as pronounced. Regardless of whether they are in urban or rural areas, speaking a dialect is more beneficial for entrepreneurial families in terms of integrating into the local community, leading to better entrepreneurial performance.

Our most insightful finding pertains to the relationship between dialects and family entrepreneurship. Families that speak Standard Mandarin are more likely to initiate businesses. As the official language of China, Standard Mandarin facilitates the acquisition of individual competencies and social networks, making it easier for entrepreneurs to access resources and information. On the other hand, families that speak dialects exhibit better entrepreneurial performance. Most family-based

entrepreneurial businesses are small-scale, with customer bases primarily consisting of residents. Speaking a dialect helps entrepreneurs gain trust and integrate more easily into the local community, thereby achieving better entrepreneurial outcomes.

From the perspective of urban and rural sub-samples, the impact of Standard Mandarin on urban families' entrepreneurship choice is more significant, but not as pronounced for rural families. For both urban and rural families, dialects are more helpful in integrating into the local community, leading to better entrepreneurial performance. From a mechanistic standpoint, cognitive and interpersonal competencies are two critical channels through which language influences family entrepreneurship choice. The use of Standard Mandarin helps enhance the formation of human capital and social capital within families, thereby increasing the probability of family entrepreneurship.

To ensure the robustness of our empirical results, we employed instrumental variable methods to address potential endogeneity issues that may arise from bidirectional causality or omitted variable bias in the influence of familial daily language (Standard Mandarin or dialect) on family entrepreneurial behavior. The results indicate that the influence of familial daily language on family entrepreneurial performance is endogenous. When controlling for endogeneity, the entrepreneurial performance of families using Standard Mandarin is not as favorable, while that of families using dialects is better, with no significant differences in the regression results.

Given the presence of ten distinct dialect areas in China, communication barriers exist between them. In most dialect areas, the use of Standard Mandarin or a dialect as familial daily language has a significant positive impact on family entrepreneurship choice, but a significant negative impact on family entrepreneurial performance. Further analysis reveals that families speaking Standard Mandarin, characterized by high income, youth, and residing in eastern and central regions, are more likely to start businesses. However, in most sub-samples, the use of familial daily language has a significant negative impact on family entrepreneurial performance, suggesting that the use of dialects helps improve the entrepreneurial performance of families.

Based on the research findings presented in this article, several insights can be derived to enhance family entrepreneurship and entrepreneurial performance in China:

Emphasis on Community Relationship Building: It is crucial to prioritize the construction of neighborhood relationships and foster a harmonious community environment. Grassroots organizations can organize a series of cultural activities to build communication platforms, thereby strengthening community ties and creating a cohesive community atmosphere. Rural families should expand their social networks based on kinship and geography, seeking higher-quality networks that broaden access to information and alleviate issues of information asymmetry. Urban families, while developing "weak ties" networks mainly grounded in "friendship and business connections," should also focus on enhancing the strength of individual network constructions, deepening the layers of social networks, and obtaining more emotional support.

Optimization of Educational Resource Allocation and Strengthening Language Education: There should be an emphasis on optimizing the allocation of educational resources and reinforcing language education. This can be achieved by providing more skill training for entrepreneurs through subsidies for education, employment training, and other means, which enhances individual literacy and improves human capital. Additionally, promoting Mandarin proficiency adapts to the higher-level language requirements in the entrepreneurial process, mitigates the adverse effects of language barriers, ensures smooth information communication, and gains recognition and identity within the community.

Promoting Balanced Urban-Rural Development and Accelerating Factor Flow: To bridge the urban-rural divide, it is necessary to promote balanced development and accelerate the flow of factors between urban and rural areas. On one hand, the government can increase investment in public resources in rural areas and provide more subsidies for innovative and entrepreneurial activities, such as vigorously supporting new types of agricultural management entities and leading enterprises. On the other hand, guiding more urban capital into rural areas, such as by supporting the development of rural tourism and sightseeing projects, can stimulate rural economies. Establishing a robust system for the balanced development of urban and rural public services and promoting equal access to education can further revitalize rural economies, support education, and achieve integrated urban-rural development by attracting investments, building characteristic towns, and supporting educational initiatives.

These recommendations aim to improve the overall environment for family entrepreneurship in China from multiple perspectives, including social relations, personal capability enhancement, and equitable resource distribution, ultimately leading to better entrepreneurial outcomes.

## Conclusion

From a linguistic perspective, this study has enriched the research on the influence of diversity on family behavior by theoretically and empirically documenting the impact mechanism of Mandarin versus regional dialects on family entrepreneurship. According to the study results, families who use Mandarin have a greater likelihood of initiating a business due to language's impact on family entrepreneurial choices, with cognitive ability and social networks serving as crucial pathways. Specifically, the use of Mandarin has a more pronounced impact on the entrepreneurial choices of urban families, but not on rural families. Conversely, families who use dialects exhibit better entrepreneurial performance, both in urban and rural areas. Dialects are more beneficial for entrepreneurial families in integrating into local social networks and achieving superior entrepreneurial performance. This article also investigates the differences in the impact of dialects on family entrepreneurship across various contexts. Our study demonstrates that the category of dialect, income level, and age level all have a moderating effect on the influence of dialects on family entrepreneurship, with variations observed between the Eastern and Central-Western regions of China.

Inevitably, this study also has limitations and shortcomings. Firstly, It is known that many study subjects can speak both Mandarin and dialects at the same time. From a typological perspective of language ability, the characterization of daily language use should be three categories: (weak Mandarin, strong dialect), (strong Mandarin, weak dialect), and (strong Mandarin, strong dialect) ; From a typological perspective of language use habits, the characterization of daily language use should be two categories: (weak Mandarin, strong dialect), and (strong Mandarin, weak dialect). The distinction between language ability and language habits is crucial for understanding the complex interplay between linguistic factors and entrepreneurial outcomes. While language ability reflects an individual's competence in using a language, language habits reveal how that ability is actually put into practice in everyday life. By considering both dimensions, we can gain a more comprehensive understanding of how language influences family entrepreneurship in China. Due to data limitations, this study adopts language usage habits as the primary research classification. This approach allows us to explore the practical implications of language choices in everyday life, which are particularly relevant to family business dynamics and social interactions. If more detailed data become available in the future, we hope to conduct further research that incorporates both language ability and language habits to provide a more nuanced analysis. Secondly, this empirical study primarily utilizes micro data from the 2018 China Family Panel Studies (CFPS). Given the rapid pace of social change in China, this data may not fully and accurately represent the current dynamics of family entrepreneurship in the country.

The impact of language on economic performance remains an emerging research field. Beyond the focus on family entrepreneurship in this study, linguistic diversity may also affect internal communication, coordination costs, and teamwork efficiency within enterprises, thereby influencing management costs and productivity levels. Investigating these issues will help further clarify the channels and mechanisms through which culture influences economic development.

## Supporting information

**S1 Data. Response to reviewers.**
(RAR)

## Acknowledgments

We would like to express our sincere gratitude to the China Survey Data Database of Peking University for providing the China Family Panel Studies (CFPS) microdata, which enabled us to conduct this study. The CFPS dataset is a valuable resource for further research on the impact of language on family entrepreneurship and related topics.

## Author contributions

**Conceptualization:** Chenglin Ren, Yijia Tang.

**Data curation:** Chenglin Ren.

**Formal analysis:** Chenglin Ren.

**Funding acquisition:** Chenglin Ren.

**Investigation:** Chenglin Ren.

**Methodology:** Chenglin Ren.

**Project administration:** Chenglin Ren.

**Resources:** Chenglin Ren.

**Software:** Chenglin Ren.

**Supervision:** Chenglin Ren.

**Validation:** Chenglin Ren.

**Visualization:** Chenglin Ren.

**Writing – original draft:** Chenglin Ren.

**Writing – review & editing:** Chenglin Ren, Yijia Tang.

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
