## [Decision Letter · Decision Letter 0]

6 Aug 2024

PONE-D-24-22860Language and Family Entrepreneurship: Empirical research based on micro survey data of CFPS in ChinaPLOS ONE

Dear Dr. Ren,

Thank you for submitting your manuscript to PLOS ONE. After careful consideration, we feel that it has merit but does not fully meet PLOS ONE’s publication criteria as it currently stands. Therefore, we invite you to submit a revised version of the manuscript that addresses the points raised during the review process.

We look forward to receiving your revised manuscript.

Kind regards,

Xiaoguang Fan

Academic Editor

PLOS ONE

Reviewers' comments:

Reviewer's Responses to Questions

**Comments to the Author**

1. Is the manuscript technically sound, and do the data support the conclusions?

Reviewer #1: Yes

Reviewer #2: Yes

2. Has the statistical analysis been performed appropriately and rigorously? 

Reviewer #1: Yes

Reviewer #2: Yes

3. Have the authors made all data underlying the findings in their manuscript fully available?

Reviewer #1: Yes

Reviewer #2: Yes

4. Is the manuscript presented in an intelligible fashion and written in standard English?

Reviewer #1: Yes

Reviewer #2: Yes

5. Review Comments to the Author

Reviewer #1: While the paper excels in many areas, the conclusion could be strengthened. It should include a more comprehensive discussion of the practical implications and contributions of the study. Additionally, it is important to acknowledge the study's limitations, particularly regarding sample size, and to provide suggestions for future research. Addressing these aspects will enhance the overall impact and applicability of the research.

Overall, this manuscript presents a well-conceived and executed study with significant potential to contribute to our understanding of the intersection between language and family entrepreneurship in China. I recommend that the authors address the noted weaknesses in the conclusion to further improve the manuscript.

Best regards,

Reviewer #2: Language and Family Entrepreneurship, a very interesting research topic. The theoretical hypotheses and empirical tests throughout the article are very adequate and scientific.

However, the following comments would hopefully help to improve the quality of the article.

The introduction section needs to clarify the research question(s) of the paper and emphasise the significance of this research topic.

It is recommended that a discussion section be included before the conclusion section to provide an in-depth account of the theoretical contributions, and practical implications of your research. As you know, the practical implications I saw in your conclusion section, and you can put it in the added discussion section.

It is necessary to add in the conclusion section the research limitations of the article as well as possible future research directions.

Good luck!

6. PLOS authors have the option to publish the peer review history of their article (what does this mean? ). If published, this will include your full peer review and any attached files.

**Do you want your identity to be public for this peer review?** For information about this choice, including consent withdrawal, please see our Privacy Policy .

Reviewer #1: **Yes: ** Lukman Raimi

Reviewer #2: No

---

## [Author Response · Author response to Decision Letter 1]

30 Aug 2024

About Reviewers' comments:

1. Is the manuscript technically sound, and do the data support the conclusions?

Reviewer #1: Yes

Reviewer #2: Yes

Response: Thanks.

2. Has the statistical analysis been performed appropriately and rigorously?

Reviewer #1: Yes

Reviewer #2: Yes

Response: Thanks.

3. Have the authors made all data underlying the findings in their manuscript fully available?

The PLOS Data policy requires authors to make all data underlying the findings described in their manuscript fully available without restriction, with rare exception (please refer to the Data Availability Statement in the manuscript PDF file). The data should be provided as part of the manuscript or its supporting information, or deposited to a public repository. For example, in addition to summary statistics, the data points behind means, medians and variance measures should be available. If there are restrictions on publicly sharing data & mdash; e.g. participant privacy or use of data from a third party & mdash; those must be specified.

Reviewer #1: Yes

Reviewer #2: Yes

Response: Thanks.

4. Is the manuscript presented in an intelligible fashion and written in standard English?

Reviewer #1: Yes

Reviewer #2: Yes

Response: Thanks.

5. Review Comments to the Author

Reviewer #1: While the paper excels in many areas, the conclusion could be strengthened. It should include a more comprehensive discussion of the practical implications and contributions of the study. Additionally, it is important to acknowledge the study's limitations, particularly regarding sample size, and to provide suggestions for future research. Addressing these aspects will enhance the overall impact and applicability of the research.

Overall, this manuscript presents a well-conceived and executed study with significant potential to contribute to our understanding of the intersection between language and family entrepreneurship in China. I recommend that the authors address the noted weaknesses in the conclusion to further improve the manuscript.

Best regards,

Response: Thanks for the reviewer's comments. We have made revisions to the conclusion section in three aspects: firstly, we have provided a more detailed explanation of the contribution and practical impact of this study; Secondly, it pointed out the limitations of this study, especially in terms of sample size; Thirdly, it pointed out the further research directions of this study and discussed other aspects of the impact of language on economic performance.

Reviewer #2: Language and Family Entrepreneurship, a very interesting research topic. The theoretical hypotheses and empirical tests throughout the article are very adequate and scientific.

However, the following comments would hopefully help to improve the quality of the article.

The introduction section needs to clarify the research question(s) of the paper and emphasise the significance of this research topic.

It is recommended that a discussion section be included before the conclusion section to provide an in-depth account of the theoretical contributions, and practical implications of your research. As you know, the practical implications I saw in your conclusion section, and you can put it in the added discussion section.

It is necessary to add in the conclusion section the research limitations of the article as well as possible future research directions.

Good luck!

Response: Thanks for the reviewer's comments. We have addressed the comments as follows:

Introduction Section: We have revised the introduction section to strengthen the exposition of the theme and importance of this study, providing a clearer context and rationale for our research questions.

Discussion Section: We have added a discussion section before the conclusion to provide a detailed explanation of the research contribution and practical impact of this study, highlighting the significance of our findings.

Conclusion Section: We have made revisions to the conclusion section to clarify the limitations of this study, particularly noting the sample size and its potential impact on the generalizability of our findings. Additionally, we have outlined further research directions to explore other aspects of the impact of language on economic performance.

6. PLOS authors have the option to publish the peer review history of their article (what does this mean?). If published, this will include your full peer review and any attached files.

Do you want your identity to be public for this peer review? For information about this choice, including consent withdrawal, please see our Privacy Policy.

Reviewer #1:Yes:Lukman Raimi

Reviewer #2: No

Response: Thanks.

About Journal requirements:

Response: Thanks for the editor's reminder. We have made revisions in the manuscript of “the article title” and “the main body” according to the format requirements of PLOS ONE's style guidelines.

Response: Thanks for the editor's reminder. We are pleased to provide our research data on this topic.

3. Please review your reference list to ensure that it is complete and correct. If you have cited papers that have been retracted, please include the rationale for doing so in the manuscript text, or remove these references and replace them with relevant current references. Any changes to the reference list should be mentioned in the rebuttal letter that accompanies your revised manuscript. If you need to cite a retracted article, indicate the article & rsquo;s retracted status in the References list and also include a citation and full reference for the retraction notice.

Response: Thanks for the editor's reminder. We have revised the references according to the format requirements of PLOS ONE's style guidelines.

---

## [Decision Letter · Decision Letter 1]

19 Nov 2024

PONE-D-24-22860R1Language and family entrepreneurship: empirical research based on micro survey data of CFPS in ChinaPLOS ONE

Dear Dr. Ren,

Thank you for submitting your manuscript to PLOS ONE. After careful consideration, we feel that it has merit but does not fully meet PLOS ONE’s publication criteria as it currently stands. Therefore, we invite you to submit a revised version of the manuscript that addresses the points raised during the review process. After reading your manuscript and the comments from anonymous reviewers, you need to make minor revisions. Please revise according to the reviewers' suggestions and provide corresponding responses. If everything goes smoothly, we will proceed to the next round of review based on your updating.

We look forward to receiving your revised manuscript.

Kind regards,

Xiaoguang Fan

Academic Editor

PLOS ONE

Journal Requirements:

Reviewers' comments:

Reviewer's Responses to Questions

**Comments to the Author**

1. If the authors have adequately addressed your comments raised in a previous round of review and you feel that this manuscript is now acceptable for publication, you may indicate that here to bypass the “Comments to the Author” section, enter your conflict of interest statement in the “Confidential to Editor” section, and submit your "Accept" recommendation.

Reviewer #2: (No Response)

Reviewer #3: (No Response)

2. Is the manuscript technically sound, and do the data support the conclusions?

Reviewer #2: Yes

Reviewer #3: Yes

3. Has the statistical analysis been performed appropriately and rigorously? 

Reviewer #2: Yes

Reviewer #3: Yes

4. Have the authors made all data underlying the findings in their manuscript fully available?

Reviewer #2: Yes

Reviewer #3: Yes

5. Is the manuscript presented in an intelligible fashion and written in standard English?

Reviewer #2: Yes

Reviewer #3: Yes

6. Review Comments to the Author

Reviewer #2: It is clearly evident that the authors ought to incorporate a more comprehensive analysis of the practical implications within the discussion section. This enhancement would significantly strengthen the overall impact and applicability of their research findings.

Reviewer #3: The manuscript under review utilizes CFPS data to explore the relationship between language and entrepreneurial behavior. The topic is innovative, the research design is rational, the methods are appropriately applied, and the conclusions drawn are credible. In the pursuit of academic rigor, the following suggestions are offered for the author's consideration to further refine the paper:

The manuscript focuses on the mechanism by which language influences entrepreneurial behavior and performance. The Sobel-Goodman test is used to examine the mediating roles of cognitive ability and social networks. It is noted that this model is typically used to test the effect of a single mediating variable. It is suggested that the author consider employing a method that allows for the comparison of multiple mediating variables, such as the KHB model. This approach could be beneficial in comparing the relative strengths of different mediators when multiple mediators are present. As the author mentions, human capital and social capital are two significant factors influencing family entrepreneurship. Comparing these two mechanisms within the same framework could potentially enhance the marginal contributions of the study.

The paper acknowledges that due to data limitations, speakers of both Mandarin and dialects are not included in the analysis. This highlights the distinction between the concepts of "language ability" and "language use habits." From a typological perspective of language ability, there should be four categories: (weak Mandarin, weak dialect), (weak Mandarin, strong dialect), (strong Mandarin, weak dialect), and (strong Mandarin, strong dialect). However, the paper only addresses two of these categories. If the data does not permit coverage of all such categories, it would be beneficial for the author to discuss the similarities and differences between these concepts in an appropriate section of the paper.

It is hoped that these suggestions will be helpful in improving the manuscript.

Best regards

7. PLOS authors have the option to publish the peer review history of their article (what does this mean? ). If published, this will include your full peer review and any attached files.

**Do you want your identity to be public for this peer review?** For information about this choice, including consent withdrawal, please see our Privacy Policy .

Reviewer #2: No

Reviewer #3: **Yes: ** Xinyan Xie

---

## [Author Response · Author response to Decision Letter 2]

16 Dec 2024

About Journal requirements:

Response: Thanks for the editor's reminder. We have thoroughly reviewed and reorganized the references in this article to ensure their completeness, accuracy, and current relevance. All cited references have been verified to confirm that they are not among those that have been retracted.

About Reviewers' comments:

1. If the authors have adequately addressed your comments raised in a previous round of review and you feel that this manuscript is now acceptable for publication, you may indicate that here to bypass the “Comments to the Author” section, enter your conflict of interest statement in the “Confidential to Editor” section, and submit your "Accept" recommendation.

Reviewer #2: (No Response)

Reviewer #3: (No Response)

Response: Thanks.

2. Is the manuscript technically sound, and do the data support the conclusions?

Reviewer #2: Yes

Reviewer #3: Yes

Response: Thanks.

3. Has the statistical analysis been performed appropriately and rigorously?

Reviewer #2: Yes

Reviewer #3: Yes

Response: Thanks.

4. Have the authors made all data underlying the findings in their manuscript fully available?

The PLOS Data policy requires authors to make all data underlying the findings described in their manuscript fully available without restriction, with rare exception (please refer to the Data Availability Statement in the manuscript PDF file). The data should be provided as part of the manuscript or its supporting information, or deposited to a public repository. For example, in addition to summary statistics, the data points behind means, medians and variance measures should be available. If there are restrictions on publicly sharing data & mdash e.g. participant privacy or use of data from a third party & mdash ; those must be specified.

Reviewer #2: Yes

Reviewer #3: Yes

Response: Thanks.

5. Is the manuscript presented in an intelligible fashion and written in standard English?

Reviewer #2: Yes

Reviewer #3: Yes

Response: Thanks.

6. Review Comments to the Author

Reviewer #2: It is clearly evident that the authors ought to incorporate a more comprehensive analysis of the practical implications within the discussion section. This enhancement would significantly strengthen the overall impact and applicability of their research findings.

Response: Thanks for the reviewer's comments. In the discussion section of the article, we delve deeper into the implications of our research findings and provide actionable recommendations aimed at enhancing the environment for family entrepreneurship in China. Our recommendations address multiple dimensions, including the cultivation of social relationships, the enhancement of personal capabilities, and the promotion of equitable resource distribution. By addressing these areas, we aim to foster more favorable conditions for entrepreneurship, ultimately leading to improved entrepreneurial outcomes and sustainable business growth.

We are grateful for your valuable suggestions, which have been instrumental in refining our research. These modifications enhance the robustness of our analysis and contribute to a deeper understanding of the factors influencing family entrepreneurship in China.

Reviewer #3: The manuscript under review utilizes CFPS data to explore the relationship between language and entrepreneurial behavior. The topic is innovative, the research design is rational, the methods are appropriately applied, and the conclusions drawn are credible. In the pursuit of academic rigor, the following suggestions are offered for the author's consideration to further refine the paper:

The manuscript focuses on the mechanism by which language influences entrepreneurial behavior and performance. The Sobel-Goodman test is used to examine the mediating roles of cognitive ability and social networks. It is noted that this model is typically used to test the effect of a single mediating variable. It is suggested that the author consider employing a method that allows for the comparison of multiple mediating variables, such as the KHB model. This approach could be beneficial in comparing the relative strengths of different mediators when multiple mediators are present. As the author mentions, human capital and social capital are two significant factors influencing family entrepreneurship. Comparing these two mechanisms within the same framework could potentially enhance the marginal contributions of the study. The paper acknowledges that due to data limitations, speakers of both Mandarin and dialects are not included in the analysis. This highlights the distinction between the concepts of "language ability" and "language use habits." From a typological perspective of language ability, there should be four categories: (weak Mandarin, weak dialect), (weak Mandarin, strong dialect), (strong Mandarin, weak dialect), and (strong Mandarin, strong dialect). However, the paper only addresses two of these categories. If the data does not permit coverage of all such categories, it would be beneficial for the author to discuss the similarities and differences between these concepts in an appropriate section of the paper.

It is hoped that these suggestions will be helpful in improving the manuscript.

Best regards

Response: Thanks for the reviewer's comments. Modifications Based on Your Feedback.

Firstly, In response to your feedback, we have adopted the Karlson, Holm, and Breen (KHB) mediation analysis method to systematically decompose and statistically evaluate the pathways through which family daily language influences family entrepreneurial behavior. Building upon the basic model, we introduced two mediating variables: social network (social) and cognitive ability (cog). This allowed us to compare and analyze the impact and contribution strength of these mechanisms within a unified analytical framework.

Secondly, We have carefully distinguished between "language ability" and "language habits." After thorough consideration, we propose categorizing language ability into three distinct groups:(Weak Mandarin, Strong Dialect), (Strong Mandarin, Weak Dialect), (Strong Mandarin, Strong Dialect). For language habits, we suggest a binary classification: (Weak Mandarin, Strong Dialect) and (Strong Mandarin, Weak Dialect). The distinction between language proficiency and language habits is crucial for understanding the complex interplay between linguistic factors and entrepreneurial outcomes. While language ability reflects an individual's competence in using a language, language habits reveal how this ability is actually put into practice in daily life. By considering these two dimensions, we can gain a more comprehensive understanding of how language influences family entrepreneurship in China.

Finally, Due to data limitations, this study adopts language usage habits as the primary research classification. This approach allows us to explore the practical implications of language choices in everyday life, which are particularly relevant to family business dynamics and social interactions. If more detailed data become available in the future, we hope to conduct further research that incorporates both language ability and language habits to provide a more nuanced analysis.

We are grateful for your valuable suggestions, which have been instrumental in refining our research. These modifications enhance the robustness of our analysis and contribute to a deeper understanding of the factors influencing family entrepreneurship in China.

7. PLOS authors have the option to publish the peer review history of their article (

what does this mean?). If published, this will include your full peer review and any attached files. If you choose “no”, your identity will remain anonymous but your reviewmay still be made public.

Do you want your identity to be public for this peer review?

For information about this choice, including consent withdrawal, please see our Privacy Policy。

Reviewer #2: No.

Reviewer #3:

Yes: Xinyan XieResponse: Thanks.

---

## [Decision Letter · Decision Letter 2]

13 Jan 2025

Language and family entrepreneurship: empirical research based on micro survey data of CFPS in China

PONE-D-24-22860R2

Dear Dr. Ren,

We’re pleased to inform you that your manuscript has been judged scientifically suitable for publication and will be formally accepted for publication once it meets all outstanding technical requirements.

Kind regards,

Xiaoguang Fan

Academic Editor

PLOS ONE

Additional Editor Comments (optional):

Reviewers' comments:

Reviewer's Responses to Questions

**Comments to the Author**

1. If the authors have adequately addressed your comments raised in a previous round of review and you feel that this manuscript is now acceptable for publication, you may indicate that here to bypass the “Comments to the Author” section, enter your conflict of interest statement in the “Confidential to Editor” section, and submit your "Accept" recommendation.

Reviewer #2: All comments have been addressed

Reviewer #3: All comments have been addressed

2. Is the manuscript technically sound, and do the data support the conclusions?

Reviewer #2: Yes

Reviewer #3: Yes

3. Has the statistical analysis been performed appropriately and rigorously? 

Reviewer #2: Yes

Reviewer #3: Yes

4. Have the authors made all data underlying the findings in their manuscript fully available?

Reviewer #2: Yes

Reviewer #3: Yes

5. Is the manuscript presented in an intelligible fashion and written in standard English?

Reviewer #2: Yes

Reviewer #3: Yes

6. Review Comments to the Author

Reviewer #2: It can be observed that the authors have made exceptionally commendable revisions. If the authors can disclose the raw data, it will benefit a wider range of scholars. Of course, it depends on the authors' considerations. Nevertheless, I strongly recommend accepting this paper. Best wish!

Reviewer #3: In the previous round of revisions, the authors have made certain modifications to the statistical model for mechanism testing and the clarification of key concepts, which have led to a more refined overall presentation of the paper. From my perspective, the paper has essentially reached a publishable standard. With the aim of further improving the quality, I offer the following suggestion to enhance the paper:

Since the paper has employed the KHB model to analyze the causal mechanism and has concluded that social ability is the primary mechanism, it is necessary for the paper to extend its discussion of this result: Why is social network a more important intermediary mechanism compared to cognitive ability? And how does such a result respond to theoretical explanations? What practical insights can be derived from it?

7. PLOS authors have the option to publish the peer review history of their article (what does this mean? ). If published, this will include your full peer review and any attached files.

**Do you want your identity to be public for this peer review?** For information about this choice, including consent withdrawal, please see our Privacy Policy .

Reviewer #2: No

Reviewer #3: **Yes: ** Xinyan Xie

---

## [Editor Report · Acceptance letter]

PONE-D-24-22860R2

PLOS ONE

Dear Dr. Ren,

I'm pleased to inform you that your manuscript has been deemed suitable for publication in PLOS ONE. Congratulations! Your manuscript is now being handed over to our production team.

Kind regards,

on behalf of

Dr. Xiaoguang Fan

Academic Editor

PLOS ONE